# The Human Monocyte—A Circulating Sensor of Infection and a Potent and Rapid Inducer of Inflammation

**DOI:** 10.3390/ijms23073890

**Published:** 2022-03-31

**Authors:** Sandra Lara, Srinivas Akula, Zhirong Fu, Anna-Karin Olsson, Sandra Kleinau, Lars Hellman

**Affiliations:** 1Department of Cell and Molecular Biology, Uppsala University, The Biomedical Center, P.O. Box 596, SE-75124 Uppsala, Sweden; sandra.lara@icm.uu.se (S.L.); fuzhirong.zju@gmail.com (Z.F.); sandra.kleinau@icm.uu.se (S.K.); 2Department of Anatomy, Physiology, and Biochemistry, Swedish University of Agricultural Sciences, P.O. Box 7011, SE-75007 Uppsala, Sweden; srinivas.akula@slu.se; 3Department of Medical Biochemistry and Microbiology, BMC, P.O. Box 582, SE-75123 Uppsala, Sweden; anna-karin.olsson@imbim.uu.se

**Keywords:** monocyte, macrophage, cytokine, chemokine, cytokine storm, LPS, inflammatory response, antigen presentation, G-CSF, IL-6

## Abstract

Monocytes were previously thought to be the precursors of all tissue macrophages but have recently been found to represent a unique population of cells, distinct from the majority of tissue macrophages. Monocytes and intestinal macrophages seem now to be the only monocyte/macrophage populations that originate primarily from adult bone marrow. To obtain a better view of the biological function of monocytes and how they differ from tissue macrophages, we have performed a quantitative analysis of its transcriptome in vivo and after in vitro stimulation with *E. coli* LPS. The monocytes rapidly responded to LPS by producing extremely high amounts of mRNA for the classical inflammatory cytokines, IL-1α, IL-1β, IL-6 and TNF-α, but almost undetectable amounts of other cytokines. IL-6 was upregulated 58,000 times, from almost undetectable levels at baseline to become one of the major transcripts already after a few hours of cultivation. The cells also showed very strong upregulation of a number of chemokines, primarily IL-8, Ccl2, Ccl3, Ccl3L3, Ccl20, Cxcl2, Cxcl3 and Cxcl4. IL-8 became the most highly expressed transcript in the monocytes already after four hours of in vitro culture in the presence of LPS. A high baseline level of MHC class II chains and marked upregulation of super oxide dismutase (SOD2), complement factor B, complement factor C3 and coagulation factor 3 (F3; tissue factor) at four hours of in vitro culture were also observed. This indicates a rapid protective response to high production of oxygen radicals, to increase complement activation and possibly also be an inducer of local coagulation. Overall, these findings give strong support for monocytes acting primarily as potent mobile sensors of infection and rapid activators of a strong inflammatory response.

## 1. Introduction

Bone-marrow-derived blood monocytes were for many years thought to be the precursor of all tissue macrophages. However, relatively recent studies have shown that most tissue macrophages originate from two early waves of cells from the yolk sac and that it is only the blood monocytes and one population of intestinal macrophages that originate from the adult bone marrow [1,2,3,4,5,6,7]. Single cell analysis of different macrophage subpopulations has also shown large differences in phenotype and therefore likely indicates tissue-specific functions [8]. Two different subpopulations of human blood monocytes have been identified, the CD14^high^ and CD16^−^ population that constitute approximately 80%, and a minor CD14 low and CD16 + population that constitutes approximately 20% of the circulating monocytes [9]. These two population show large similarities but may have partly different functions in the amount of cytokines produced and levels of MHC class II expressed [9]. In addition to the circulating monocytes, there seems to be a relatively large pool of immature monocytes, more numerous than the circulating pool, residing in the spleen as a reservoir of monocytes ready to exit the spleen and accumulate in injured tissue [10].

During inflammatory conditions, monocytes can enter tissues and become tissue macrophages and their phenotype likely depends on the tissue environment [11,12,13,14,15]. A similar situation has recently been observed for mast cells. The majority of the tissue mast cells seem to originate from an early wave of cells from the yolk sac and bone-marrow-derived mast cell precursors primarily enter tissue during inflammatory conditions such as parasite-infected intestinal regions or inflamed lungs [16,17,18]. Following the clearance of the infection, the majority of these cells disappear, most likely by apoptosis. Most of the tissue macrophages and mast cells seem therefore to have the capacity to proliferate and thereby to restore homeostatic levels of cells if they have been consumed during an inflammatory reaction. What is then the primary function of blood monocytes when they apparently take minor part in homeostatic maintenance of the majority of tissue macrophage populations?

In order to look deeper into this issue, we here present a quantitative analysis of the transcriptome of human CD14 positive blood monocytes and how they respond to bacterial lipopolysaccharides (LPS). CD14 acts as a coreceptor to Toll-like receptor 4 (TLR-4) that together with myeloid differentiation factor 2 (MD-2) is the key sensor of *Escherichia coli* LPS [19]. Most previous studies of monocytes and macrophages have been single cell analysis with lineage tracing as the primary aim, which provides very limited quantitative information. For studies aiming to clarify the biological function and relevance of cells or molecules, high-resolution quantitative information is essential. By performing transcriptome analysis of purified human blood monocytes, we can here show that human peripheral blood monocytes act as very potent and rapid activators of inflammation by quick upregulation of a highly selective set of inflammatory cytokines, such as the classical IL-1α, IL-1β, IL-6 and TNF-α, and a number of inflammatory chemokines in response to LPS. One of the most extreme upregulations was seen for IL-6, which was upregulated more than 58,000 times within four hours of in vitro culture in the presence of *Escherichia coli* LPS. A low basal expression level of IL-8 was detected, along with very strong upregulation of this chemokine already after four hours of LPS stimulation, when it became the most highly expressed gene in these monocytes. Only a few additional genes were upregulated, among them the super oxide dismutase (SOD2) that was increased by 28 times in expression levels. The complement factor B was also upregulated by more than 2500 times and coagulation factor 3, the tissue factor, by more than 7000 times, already at four hours of in vitro culture in the presence of LPS. This shows that monocytes act as extremely potent and rapid activators of an inflammatory response by producing massive amounts of a selective set of inflammatory cytokines and chemokines and a few additional proteins of importance for their role as inflammatory initiators.

## 2. Results

### 2.1. Purification of Monocytes from Human Peripheral Blood

Monocytes were purified from concentrated white blood cells, supplied as buffy coats, from five different donors, two females (ages 47 and 61) and three males (ages 51, 43 and 28). The cells from the different donors were first washed in PBS and then subjected to two-step purification using density gradient centrifugation followed by magnetic bead separation using a monoclonal anti-human CD14 antibody. Following this two-step purification protocol, we obtained on average a 95% pure population of blood monocytes from all five donors (Figure 1). Approximately 4 million of these cells from each of the five donors were immediately pelleted and total RNA was purified by a standard protocol.

### 2.2. In Vitro Culture of Purified Peripheral Blood Monocytes

The remaining cells from each of the donors were divided into different culture dishes, with approximately equal numbers of cells. For three of the donors, three culture dishes were grown in the presence of only cell culture medium. One dish was cultured for 4 h, one dish for 24 h and one for 48 h. For four of the monocyte preparations, an identical set of three dishes was cultured for 4, 24 and 48 h with the addition of 1 ug of *E. coli* LPS/mL to the culture medium. We selected a relatively high LPS concentration to ensure maximal response by the monocytes. For one additional sample, acting as a reference sample, the LPS was replaced with 200 ng/mL of human recombinant IFN-γ. Following the in vitro culturing, the cells were harvested and total RNA was prepared from each of the different cultures to be sent for transcriptomal analysis.

### 2.3. Ampliseq Analysis of the Total Transcriptome of the Different Monocyte Samples

Total RNA from the untreated cells and from the different cultures was analyzed by the Thermo Fisher Ampliseq technology. The result was delivered in the form of Excel files with the normalized reads from a total of approximately 20,800 separate transcripts. The data file was then analyzed manually to compare the number of transcripts for each sample. The results were separated into ten different categories and are presented in Appendix A. The most dramatic changes in expression levels were among the inflammatory cytokines and chemokines and a few additional genes such as SOD2 and complement factor B, which are presented in several summary figures (Figure 2, Figure 3, Figure 4 and Figure 5). During the listing of the different transcripts and their expression levels in the text below, we give the values of the first donor, the 61-year-old man, as the levels are relatively similar between the five donors. However, all the data for all the genes we discuss in this communication, from all five donors, are listed in the Appendix A.

### 2.4. Influence of In Vitro Culture on the Monocyte Transcriptome

Putting freshly isolated human monocytes in in vitro culture may have a profound effect on their transcriptome if they attach to a plastic surface. Plastic surfaces may mimic foreign particles entering the circulation in vivo. To overcome such problems, new types of culture flasks (Cellstar) have been developed to avoid adherence and thereby culture cells under conditions that as much as possible mimic the in vivo situation. When using the Cellstar culture flasks, we could observe that the absolute majority of the monocytes stayed as non-adherent cells in the cultures and did so for several days in culture. However, even if the cells remained relatively in vivo like, a few genes were still markedly upregulated and that was the case particularly after 24–48 h in culture. We saw a marked increase in the expression of apolipoprotein E (APOE), from a few reads to between 2400 and 3000 reads in these monocytes, which corresponds to an increase of almost 3000-fold by 48 h of in vitro culture. We also observed an increase in the expression of the oxidized low-density lipoprotein receptor (OLR1) by almost 1000-fold by four hours of in vitro culture (Appendix A). RNASE1 was also very strongly upregulated only by in vitro culturing from 3 to 1894 reads after 48 h in culture in one of the three cultures but only to around 200 reads in the two other cultures. A marked increase in several additional genes after 48 h in culture was also observed. C1QA went from 13 to 287 reads, and SLAMF8 went from 1 to 179 in one of the subjects. However, for the absolute majority of genes, very small changes did occur upon in vitro culture in the absence of any additional stimulant. However, and importantly, almost none of the classical inflammatory cytokines and the chemokines were upregulated by LPS, possibly except CCL2, indicating a different type of low-level induction (Appendix A).

### 2.5. Lysozyme, MHC Molecules and Pattern Recognition Receptors and Other Immune-Related Molecules

The most highly expressed gene in these monocytes directly after purification was lysozyme with 27,394 reads in the first donor. This should be compared to the housekeeping gene β-actin with around 20,000 reads in the five donors with some variation between donors (Appendix A). Lysozyme is one of the few antibacterial proteins we detect in these monocytes. We do not find any reads for other antibacterial proteins such as defensins and cathelicidin, which in general are highly expressed and stored in large amounts in cytoplasmic granules by human neutrophils.

MHC Class I and II were in the untreated cells expressed at relatively high levels, in the range of 1000–5500 reads in all of these five donors. However, we see a peculiar pattern that only some of the chains are expressed at high levels whereas others are almost undetectable. For MHC class II, it is primarily HLA-DRA, HLA-DRB1, HLA-DPA1 and HLA-DPB1 that are expressed at high levels. A low level of HLA-DQA1 was also detected whereas DPB2, DQA2 and DQB2 were almost undetectable (Appendix A). Interestingly, LPS stimulation results in a marked downregulation of the MHC class II genes (Appendix A). For MHC class I, we see high levels of the common beta chain, the β2-microglobulin, and high levels of HLA-A, but only very low, almost undetectable, levels of HLA-B and/or C in some of the donors (Appendix A). We detect high levels of the non-classical class I gene HLA-E in all the donors and this expression seems relatively independent of in vitro cultivation and both LPS and IFN-γ stimulation (Appendix A).

The pattern recognition receptors such as the Toll-like receptors (TLR) were expressed at remarkably low levels. Most of them were in the range of 20–200 reads, indicating that low numbers are sufficient for rapid activation (Appendix A). However, upon in vitro culture, the expression increases quite dramatically at least for some of the TLRs, such as TLR4 and TLR2. The transcript counts for TLR4, for example, increased from 25 to 246 in the third donor. This increase was also slightly enhanced by LPS stimulation in some of the individuals (Appendix A).

The only other immune-related transcripts where we see a marked increase in expression, excluding cytokines and chemokines, are complement factors B (CFB), complement factor C3 and coagulation factor 3 (F3) (Appendix A). CFB increased from 0.2 to 508 reads, which corresponds to approximately 2500 times upregulation already after four hours of culture in the presence of LPS, and complement factor C3 increased from 2 to 118 reads in donor 1 and coagulation factor 3 (F3), also named tissue factor, was an important initial trigger of coagulation from 0 to 717 in donor 3 (Appendix A). In contrast, two other complement components, properdin and ficolin, involved in the alternative and the mannose activation pathways, respectively, were slightly downregulated from 991 to 410 for properdin and 3198 to 849 for ficolin in the first donor (Appendix A).

A remarkable upregulation of superoxide dismutase (SOD2) was also observed. Baseline levels of 1113 reads increased to 31,170 reads after 4 h of in vitro culture in the presence of LPS, which is an increase by 28 times to a level exceeding the highest transcript level of any gene, the lysozyme (27,394 reads), before activation (Appendix A and Figure 5). SOD2 is important for quenching super oxide ions, which are generated in high amounts upon activation of monocytes and macrophages by the cytochrome b558 for killing of bacteria within phagosomes. The high upregulation may here be protection against damages of the cell caused by oxygen radicals.

We also observed quite strong upregulation of several cell adhesion molecules such as integrins B3 and B8, phospholipase A2 (PLA2G7), the indole amine 2,3 dioxidase 1 (IDO1), the antiviral zinc finger ZC3H12C, the myxovirus resistance gene MX2 and the non-coding RNA, MIR155HG, upon LPS stimulation (Appendix A). MIR155HG went from 0.1 to 182 reads by four hours in the presence of LPS, indicating that it may be involved in regulating and controlling the massive increase in cytokine and chemokine mRNA induced by LPS (Appendix A).

A strong upregulation of a few proteases and protease inhibitors was also observed by LPS stimulation, such as the matrix metalloprotease 14 (MMP14), the protease inhibitors cystatin B2 and B9 and the peptidase inhibitor 3 (PI3) (Appendix A).

For the different CD molecules, we find a 3–5-fold upregulation of CD14 by 24–48 h of LPS stimulation (Appendix A). In contrast, CD4 decreased almost 50-fold in some of the donor monocytes after 24–48 h (Appendix A). The expression of B7:2 (CD86) decreased after four hours LPS stimulation by 4–20-fold, whereas B7:1 (CD80) increased by 100–500-fold, from 2 to 318 in donor 1 (Appendix A). As expected, we did not detect any expression of marker for immature hematopoietic cells, the CD34, or the B cell marker CD19 on these cells (Appendix A).

### 2.6. Fc Receptors

Fc receptors for the different immunoglobulins are important for the uptake of microbes and other foreign particles in the circulation by phagocytosis. The expression of both IgG and IgA receptors was detected. However, the steady state levels of the IgG receptors seem to differ markedly between individuals. Among these five donors, we can observe a more than ten-fold difference between individuals, from 55 reads for FcRγ2A in one individual to 756 in another (Appendix A). However, the expression levels do not change markedly during in vitro culture (Appendix A). Low levels of Fc-gamma receptor 3A and the high-affinity receptor for IgG FCGR1A were observed but there was almost no FCGR3B (Appendix A). The FCGR3A was markedly downregulated by LPS (Appendix A). In contrast, the IgA receptor (FCAR or CD89), which was expressed at relatively low levels before activation, was instead upregulated by LPS (Appendix A). Interestingly, there were also very low levels of both the high and low affinity IgE receptors, the IGER1A and FCER2, contesting the earlier studies of expression of both of these two receptors on human monocytes (Appendix A).

### 2.7. Cytokines

Relatively modest changes were seen for most of the molecules described above compared to what we observed for a limited set of cytokines, primarily or almost exclusively the classical inflammatory cytokines, IL-1α, IL-1β, IL-6 and TNF-α (Appendix A and Figure 2). The most extreme was the upregulation of IL-6 where the levels go from steady state levels of almost undetectable 0.1 reads to 5850 reads, which is an upregulation by more than 58,000 times after only four hours of in vitro culture in the presence of LPS (Appendix A and Figure 2). The other extreme was IL-1β, which went from 11 to 31,674 reads, which is a level well above the highest baseline transcript, lysozyme, with its 27,394 reads (Appendix A and Figure 2). IL-1-α went from 0.2 to 2134 reads after four hours in culture, corresponding to an increase of more than 13,000 times. TNF-α went from a relatively high baseline of 131 to 2047 and a 16-times increase in expression already after four hours in culture (Appendix A and Figure 2). Interestingly, there was also almost a complete absence of most of these different cytokines at baseline, before activation, and extremely high levels after only four hours of LPS stimulation (Appendix A and Figure 2). We also observed a strong upregulation of G-CSF (CSF3) by LPS. This cytokine is of major importance for recruitment and activation of neutrophils, which is why the rapid and potent increase in this cytokine by LPS, from 0 to over 700 reads, can be of major importance upon a bacterial infection (Appendix A).

Due to the large amounts of inflammatory cytokines produced by the LPS-stimulated monocytes, most likely there is also a need to balance this response, so as not to cause an excessive inflammatory response. We can actually see such a tendency in the transcriptome. With a quite dramatic upregulation of the IL-1 receptor antagonist IL1RN and TNIP3 the inhibitor of an IL-1, TLR-4- and TNF-α-induced NFkB activation was observed. The IL1RN was upregulated from 20–50 reads before activation to exceeding 1000 reads after LPS stimulation, and TNIP3 was in one of the monocyte cultures upregulated from 0.2 reads before LPS induction to 1882 after 24 h in the presence of LPS (Appendix A).

### 2.8. Chemokines

A very similar picture was seen for a selective panel of chemokines. A remarkable upregulation was seen for IL-8, from 1202 to 43,405 reads at four hours. IL-8 thereby became the most highly expressed transcript in these cells after only four hours in the presence of LPS. Very high levels were also observed for Ccl2, Ccl3, CclL3, Ccl4, Ccl20, Cxcl1, Cxcl2 and Cxcl3 (Appendix A and Figure 3 and Figure 4). For some of these chemokines, the upregulation was delayed in time and the highest expression levels were seen at 24 or 48 h of in vitro culture in the presence of LPS. This was the case for Ccl2 where we observed an almost 20-fold increase between 4 and 24 h in two of the donors (Appendix A and Figure 3). A similar situation was seen for Cxcl5 where only a modest upregulation was seen at 4 h but a very strong upregulation was seen at 24 and 48 h (Appendix A and Figure 4). At 48 h, the Cxcl5 levels were at over 10,000 reads, from undetectable at baseline (Appendix A and Figure 4).

### 2.9. Receptors for Cytokines and Chemokines

A few receptors were expressed at relatively high baseline levels in these monocytes, including the IL10RA and IL17RA with expression levels in the range of 100–600 reads (Appendix A). Several other receptors were expressed at very low levels of 0–29 reads before activation such as IL15RA, IL7R, IL2RA, CCR7 and the adenosine receptor A2 (ADORA2A) (Appendix A). However, a few of these were heavily upregulated in expression after four hours of LPS stimulation (Appendix A). ADORA2A went from 14 to 666 reads in the first individual and IL2RA went from 0 to 382 in the second individual by four hours in the presence of LPS (Appendix A).

### 2.10. Effect on the Monocyte Transcriptome by Culturing the Cells in the Presence of IFN-γ

As an alternative to the LPS stimulation, representing a non-bacterial PAMP activation pathway, we analyzed the effect of culturing the monocytes from one donor in the presence of 200 ng/mL of human recombinant IFN-γ. Compared to what we observed with LPS, a very different pattern in expression was seen for these monocytes. Almost no increase in the cytokines and chemokines upregulated by LPS was observed. The major effect on these cells by the addition of IFN-γ was instead a modest upregulation of proteins involved in antigen presentation such as the alpha and beta chains of MHC Class II, DP, DR and DQ, the chains involved in transport of these chains to the endosomal compartment such as the invariant chain. We also observed a marked increase in B7:1 (CD80), a receptor of major importance for the activation of T cells during the antigen presentation process. B7:1 (CD80) went from 0.8 to 51 reads by four hours in the presence of IFN-γ (Appendix A). However, the second member of this small family of immunoregulatory proteins, B7:2 (CD86), which was relatively high at baseline before IFN-γ addition, was instead downregulated by IFN-γ addition, from 108 to 50 by four hours (Appendix A). A few cytokines and chemokines were also upregulated, but not at all as much as in the response to LPS. The pattern was also different from what we saw with LPS. The cytokines and chemokines upregulated by IFN-γ were instead IL-27, CXCL9, CXCL10 and CXCL11. CXCL9 went from 2.5 reads to 6084 after 24 h in the presence of IFN-γ, CXCL10 from 1.7 to 1554 reads by four hours and CXCL11 from 0.2 to 1494 by four hours (Appendix A). IL-27 had a more modest increase from 12 to 129 at four hours of incubation in vitro (Appendix A). The high-affinity receptor for IgG FCGR1 also increased markedly in the presence of IFN-γ (Appendix A). It went from 60 to 704 reads after 24 h, which is a more than 10-fold increase (Appendix A).

## 3. Discussion

In humans, monocytes constitute between 2 and 10% of the white blood cells and are thereby a relatively abundant immune cell of the peripheral blood. Recently, they have been found to only contribute to a very minor extent to the majority of tissue macrophage populations, so the question is, what are their major functions?

Evidence clearly shows that they can migrate to inflamed tissue, enter the tissues and become tissue macrophages of a type determined by the tissue they enter, through cell-to-cell contacts and the cytokine environment of the tissue [11,12,13,14,15]. There, they support the local macrophage population together with incoming neutrophils by phagocytosis and possibly also additional recruitment by the production of cytokines and chemokines. However, is this their primary function? The very rapid and extremely potent upregulation of a very selective panel of inflammatory cytokines and chemokines strongly supports the hypothesis that they predominantly function as sensitive detectors for the presence of bacteria in the circulation and act as potent inducers of an inflammatory response. The very restrictive set of inflammatory cytokines essentially involving only the traditional inflammatory cytokines IL-1α, IL-1β, IL-6 and TNF-α strongly supports this conclusion. What is then the role of the set of chemokines produced upon this rapid response? One of the major upregulated chemokines is IL-8, a potent chemoattractant of neutrophils, one of the key cells for combating a bacterial infection. IL-8 is also an activator of both phagocytosis, neutrophil extracellular trap (NET) formation and later also angiogenesis, and is thereby an important player in the inflammatory response. In these activated monocytes, IL-8 actually becomes the dominating transcript only four hours after activation with 47,095 reads and is thereby the most highly expressed transcript of the activated monocyte (Appendix A and Figure 3). A few additional chemokines including Ccl2, Ccl3, CclL3, Ccl4, Ccl20, Cxcl1, Cxcl2 and Cxcl3 were also very strongly upregulated already at four hours after activation (Appendix A and Figure 3 and Figure 4). With a slight delay, peaking at 24 to 48 h, there is also Cxcl5 (Appendix A and Figure 4). Ccl4, also named MIP-1β, the second most highly expressed chemokine with 20 000 reads at four hours after activation, is a potent chemoattractant for NK-cells, monocytes and a number of other inflammatory cells [20]. The third most highly expressed chemokine, Ccl3, also named MIP-1α, with 8000 reads, is also a strong chemoattractant of neutrophils but also acts on monocytes and macrophages (Appendix A and Figure 3) [20]. Ccl2 shows a modest increase in expression by four hours with 1267 reads, but then increases dramatically by 24 h to 25,206 reads, being the second highest expressed chemokine by 24 h (Appendix A and Figure 3). Ccl2 primarily attracts monocytes and basophils, indicating that activated monocytes can further enhance the response by attracting more of their own kind. This delay in response may have functional implications. If infection does not clear within 12 h, monocytes at the site of infection may need to recruit more monocytes to keep the inflammation going. This timing thereby indicates a nicely orchestrated response.

A potent upregulation of G-CSF already by four hours after LPS stimulation was also observed, indicating that monocytes do not only contribute to the inflammatory response through the classical inflammatory cytokines. In addition, they recruit neutrophils and other inflammatory cells by production of a panel of chemokines, which may contribute to triggering the bone marrow to produce more neutrophils and to activate them to be ready for phagocytosis of bacteria in the area of inflammation by producing significant amounts of G-CSF.

The rapid and potent induction of the IL-1 receptor antagonist, IL1RN and of TNIP3, the inhibitor of NFkB signaling, after activation by IL-1, TLR-4 and TNF-α, also indicates that the monocytes modulate the response to these potent inflammatory cytokines by producing receptor antagonists and potent inhibitors of the NFkB-triggered inflammatory response.

We also observed a marked upregulation of coagulation factor 3 (F3) (Appendix A). F3, also named tissue factor, is an important initial trigger of coagulation, indicating that coagulation may be part of the bacterial defense by trapping the bacteria in the area of entry by forming a local blood clot similar to the formation of extracellular traps by neutrophils.

A marked upregulation of one microRNA was also observed upon LPS stimulation of the MIR155HG, indicating its involvement in regulating the massive upregulation of cytokines and chemokines during the response to LPS. This microRNA seems to have a very complex role during inflammation, initially to suppress negative regulators of inflammation and later to enhance NFkB activation [21,22].

The response to IFN-γ was remarkably different from that to LPS. Only a relatively modest upregulation of a few cytokines and chemokines was observed, primarily IL-27, CXCL9, CXCL10 and CXCL11 (Appendix A). The most pronounced response was for CXCL11, which went from 2.5 to 6084 reads after 24 h and CXCL9, which went from 0.2 to 1494 reads by 4 h incubation in the presence of IFN-γ (Appendix A). We found, instead of a massive increase in inflammatory cytokines, an upregulation of components connected to antigen presentation such as the MHC class II alpha and beta chains, the invariant chain, the TAP peptide transporter and also the B7 molecules. The B7 molecules are essential for the triggering of a T cell response by binding to CD28. Interestingly, for both the response to IFN-γ and LPS, we see a marked shift in the expression of the two B7 molecules. Both IFN-γ and LPS stimulation results in a marked reduction in the expression of B7:2 and an upregulation of B7:1 (Appendix A). At least in some studies, B7:1 seems to be a more potent activator of T cells in stoichiometric terms, indicating that the monocytes after inflammatory signaling can become better antigen presenters to naïve T cells [23].

The non-adhesive coating of the culturing flasks has resulted in a major improvement in the culturing of blood monocytes. We observed that the absolute majority of the cells stayed non-adherent even after 24 and 48 h in culture, better mimicking the in vivo conditions compared to previous culture flasks. However, even if the cells were non-adherent, some changes in the transcriptome occurred due to culture, primarily after 24 and 48 h and not at 4 h, as for the very rapid activation of cytokines and chemokines by LPS. We did not observe any major increase in any of the inflammatory cytokines and chemokines but instead in molecules involved in lipid metabolism such as APOE and OLR1. The mechanism and importance of these changes in transcriptome are not known but need to be kept in mind using in vitro cultured monocytes in studies of their in vivo function.

One obvious question is also how well transcriptome data match with protein expression. Large combined studies of transcriptome and proteome have shown good correlation between the two, as exemplified by the study by Meissner of LPS-activated macrophages on the secretome of these cells [24]. However, there are exceptions. In the human lung, there are mast cells that express high levels of tryptase and carboxypeptidase A3 (CPA3) mRNA but where the CPA3 protein most likely is degraded in the lysosomal compartment and therefore is not granule-stored and cannot be detected upon histochemical analysis [25]. We therefore expect the massive increase in cytokine and chemokine transcripts to also result in a similar increase in secreted protein. However, there could be some discrepancies between the two, due to processing and transport.

Two studies of the effect of LPS stimulation in vivo in mice have recently also shown strong effects on both mRNA levels in inflammatory cytokines and chemokines and on protein levels in the serum of IL-6, TNF-α, IL-1β and IL-8 when using levels of LPS comparable or higher than what we use in vitro [26,27].

In summary, these data indicate that human monocytes act as a highly sensitive and very potent mobile sensor of infection. Putting the cells in culture along with the presence of bacterial LPS trigger the cells to a massive inflammatory response and the production of massive amounts of a selective panel of cytokines and chemokines, a cytokine storm. The cytokines produced trigger upregulation of an acute phase response by the liver, primarily by IL-6, to increase levels of C-reactive protein (CRP) and other complement components and serum amyloids. The TNF-α upregulates adhesion molecules on the blood vessel endothelial cell surface to increase influx of inflammatory cells, including the monocytes themselves but primarily neutrophils, and the chemokines guide the inflammatory cells into the area of infection and enhance phagocytic activity. All of these findings point in the direction that the monocyte primarily acts as a sensitive sensor and a potent amplifier of the inflammatory response to various pathogens. The response to IFN-γ was quite different, with an upregulation of a few other chemokines, primarily CXCL9, 10 and 11, almost no upregulation of cytokines, except for a minor upregulation of IL-27, and also an upregulation of proteins connected to antigen presentation such as the MHC Class II genes, the invariant chain, the TAP transporter and the B7:1 molecule. This shows that monocytes adapt the response to the type of inflammatory challenge, one highly relevant and extremely potent cytokine and chemokine response to a bacterial challenge and a completely different response to viruses, with upregulation of a small set of chemokines and of the components involved in antigen presentation.

## 4. Materials and Methods

### 4.1. Purification of Monocytes from Human Peripheral Blood

Peripheral blood monocytes were isolated from whole blood, obtained as buffy coats, from five healthy donors at the University Hospital in Uppsala, Sweden. These five donors were of different age and sex, three men of age 51, 43 and 28, and two women of age 47 and 61. Peripheral blood mononuclear cells (PBMCs) were isolated using Ficoll–Paque Plus (GE Healthcare, Uppsala, Sweden) and standard density gradient centrifugation. PBMCs were further washed with PBS containing 2 mM of EDTA, and incubated with anti-CD14-coated magnetic beads (Miltenyi Biotec, Bergisch Gladbach, Germany). Positive selection of CD14^+^ cells was performed through magnetic cell separation. Subsequently, CD14 cells were stained with anti-human CD14 PE antibody (clone: 61D3, Invitrogen, Carlsbad, CA, USA) and the purity was verified (average of 95%) by flow cytometry.

Four million of these cells were immediately frozen and stored at −80 °C for preparation of total RNA. The remaining cells were transferred into six different culture flasks with approximately 2.5 million cells per flask. We used Cellstar culture flasks with a cell-repellent surface, developed for minimal activating properties, with white filter screw cap sterile 50 mL (25 cm^2^) (Greiner Bio-One GmbH, Kremsmünster, Austria, product number 690985). Three culture flasks were used to culture cells without any immunostimulant, only in the presence of culture medium, RPMI-1640 with 10% fetal bovine serum (FBS). Three flasks were used to culture the cells with 1 ug/mL of *Escherichia coli* LPS (Sigma-Aldrich, Saint Louis, Missouri, USA, L4516- from *E. coli* O127:B8), or 200 ng/mL of recombinant human IFN-γ (Bio-Rad, Hercules, CA, USA, cat. PHP050). Cells from these cultures were harvested at three time points, 4, 24 and 48 h of in vitro culture.

### 4.2. Ampliseq Analysis of the Total Transcriptome

Total RNA was prepared from the CD14^+^ monocytes, both the freshly isolated and the different in vitro cultures from each donor, using the RNeasy Plus mini kit (Qiagen, Hilden, Germany), according to the manufacturer’s recommendations. The RNA was eluted with 30 μL of DEPC-treated water, and the concentration of RNA was determined by using a Nanodrop ND-1000 (Nano Drop Technologies, Wilmington, DE, USA). Later, the integrity of the RNA was confirmed by visualization on 1.2% agarose gel using ethidium bromide staining.

The transcriptome of freshly isolated monocytes and the different cultures were analyzed for their total transcriptome by the Thermo Fisher chip-based Ampliseq transcriptomic platform at the SciLife lab in Uppsala, Sweden (Ion-Torrent next-generation sequencing system). The sequence results were delivered in the form of Excel files with normalized expression levels for an easy comparison between samples.

## Figures and Tables

**Figure 1 ijms-23-03890-f001:**
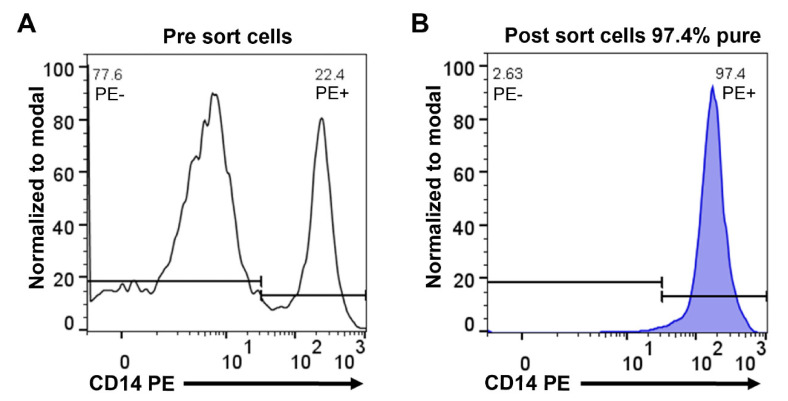
Purity of human peripheral blood monocytes obtained from PBMCs by magnetic cell sorting using CD14 microbeads. Separated cells were stained with anti-human CD14 phycoerythrin (PE) antibody and analyzed by flow cytometry. Representative flow cytometry histograms show PBMCs before sorting (**A**) and cells after sorting (**B**).

**Figure 2 ijms-23-03890-f002:**
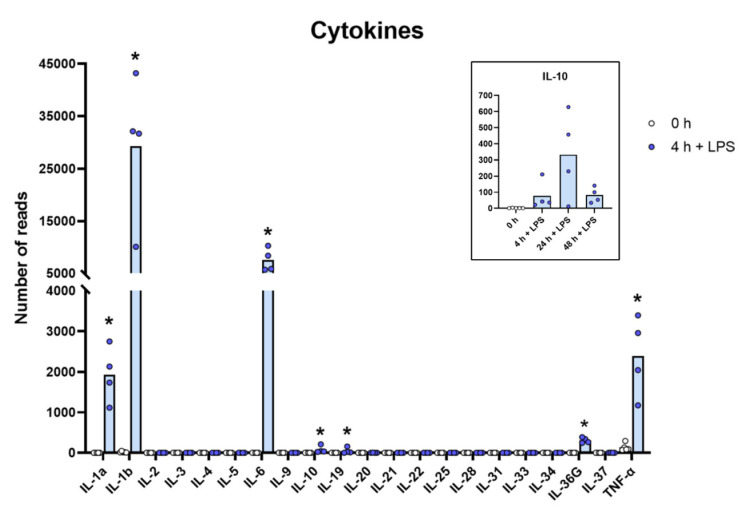
The expression levels of a panel of cytokine genes are presented to highlight the massive increase in the classical inflammatory cytokines IL-1α, IL-1β, TNF-α and IL-6. The individual values in reads and the average of the four LPS samples and the five non-induced samples are marked in the figure as individual dots. All of the actual values (reads) for all of the genes included in the figure can be found in the Appendix A. Only values for the 0 and the 4 h time points are presented in the main figure except for the values for IL-10, which are inserted as a separate panel within the figure where values for 24 and 48 h are also included. Asterisks indicate statistical differences between untreated (0 h) and treated cells with LPS (4 h) determined by a two-tailed Mann–Whitney U-test using GraphPad Prism 8 software (version 8.4.2) * *p* < 0.05.

**Figure 3 ijms-23-03890-f003:**
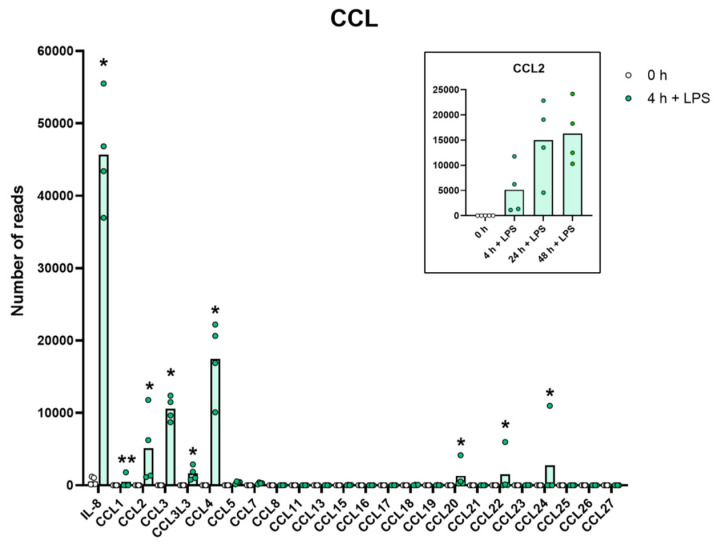
The expression levels of a panel of CCL chemokine genes are presented to highlight the massive increase in some of the classical inflammatory chemokines. The individual values and the average of the four LPS samples and the five non-induced samples are marked in the figure as individual dots. All of the actual values for all of the genes included in the figure can be found in the Appendix A. Only values for the 0 and the 4 h time points are presented in this figure except for the values for Ccl2 which are inserted as a separate panel within the figure where values for 24 and 48 h are also included. Asterisks indicate statistical differences between untreated (0 h) and treated cells with LPS (4 h) determined by a two-tailed Mann–Whitney U-test using GraphPad Prism 8 software (version 8.4.2) * *p* < 0.05, ** *p* < 0.01.

**Figure 4 ijms-23-03890-f004:**
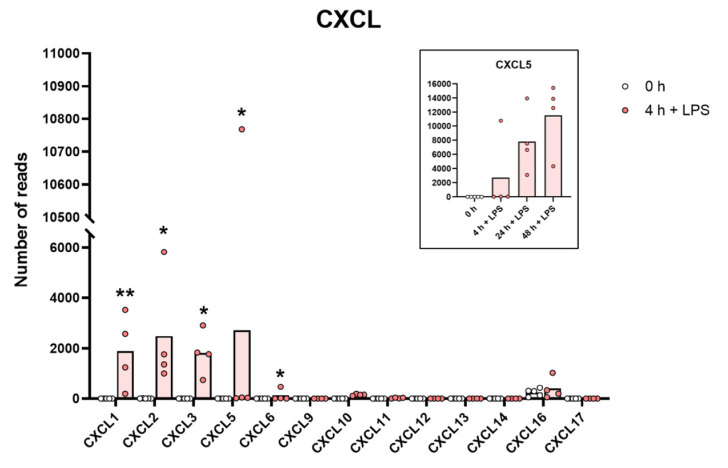
The expression levels of a panel of CXCL chemokine genes are presented to highlight the massive increase in some of the classical inflammatory chemokines. The individual values and the average of the four LPS samples and the five non-induced samples are marked in the figure as individual dots. All of the actual values for all of the genes included in the figure can be found in the Appendix A. Only values for the 0 and the 4 h time points are presented in this figure except for the values for Cxcl5 which are inserted as a separate panel within the figure where values for 24 and 48 h are also included. Asterisks indicate statistical differences between untreated (0 h) and treated cells with LPS (4 h) determined by a two-tailed Mann–Whitney U-test using GraphPad Prism 8 software (version 8.4.2) * *p* < 0.05, ** *p* < 0.01.

**Figure 5 ijms-23-03890-f005:**
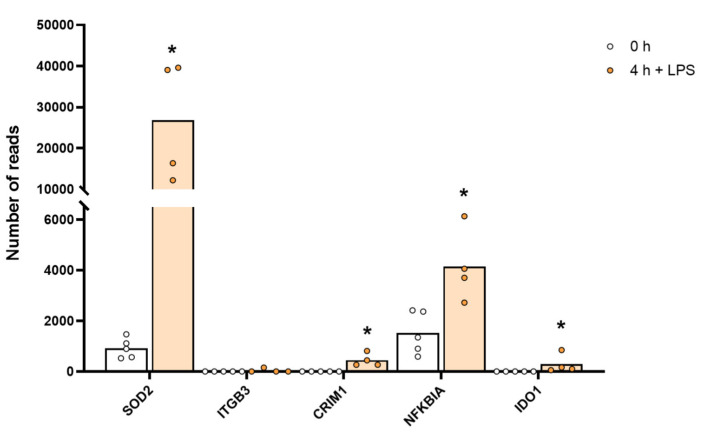
The expression levels of a few selected non-cytokine and chemokine proteins that were markedly upregulated by LPS stimulation, including SOD2. The individual values and the average of the four LPS samples and the five non-induced samples are marked in the figure as individual dots. All of the actual values for all of the genes included in the figure can be found in Appendix A. Only values for the 0 and the 4 h time points are presented. Asterisks indicate statistical differences between untreated (0 h) and treated cells with LPS (4 h) determined by a two-tailed Mann–Whitney U-test using GraphPad Prism 8 software (version 8.4.2) * *p* < 0.05.

## Data Availability

All the data of importance for this article is found in Appendix A.

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
