# Peer review of "The Human Monocyte—A Circulating Sensor of Infection and a Potent and Rapid Inducer of Inflammation"

_ijms, 2022, doi:10.3390/ijms23073890_

Round 1

Reviewer 1 Report

Manuscript ID: ijms- 1654882

Journal: International Journal of Molecular Sciences

Title: The human monocyte – a circulating sensor of infection and a 2 potent and rapid inducer of inflammation

The authors draw attention to the importance of biological function of monocytes and how they differ from tissue macrophages. The authors have performed transcriptome analysis of purified human blood monocytes in which act as very potent and rapid activators of inflammation by quick  upregulation of a highly selective set of inflammatory cytokines, such as the classical IL- 8 1a, IL-1b, IL-6 and TNF-a, and a number of inflammatory chemokines in response to LPS. One of the most extreme upregulations was seen for IL- 6, which was upregulated more than 58 000 times within four hours of in vitro culture in the presence of Escherichia coli LPS.

The paper is well organized and well written and represents a contribution to the field of the inflammation controlled by monocytes.  I personally recommend the publication of this work.

I have only few suggestions: the authors could better organize the introduction, making it legible for an article than for a review. Therefore, the introduction is too long.

Author Response

Dear Reviewer- Thanks for kind comments. Concerning the reduction in size of the introduction I am sorry to say that I have difficult to see how we could make it shorter as it already is one of the shortest introductions we have produced and we see no such comments by reviewers 2 and 3. We need to introduce the subject by a short section and then give a short teaser of the results to come.

Reviewer 2 Report

General comments:

In this work the authors present analytical transcriptome data of (bone-marrow-derived) cultured human monocytes, stimulated with LPS, to demonstrate the expression profiles of a variety of mRNAs for cytokines, chemokines and other factors involved in inflammation and immune defense. They isolated the CD14-high monocyte population and present 11 tables in which they listed the x-fold generated reads of each mRNA. Although the data are not that surprising or novel, they provide a comprehensive mRNA expression data set for comparative purposes and future studies. Several other issues need to be clarified and critically discussed, and a proper check of syntax and style is required.

Specific comments:

  1. Introduction/Discussion: The authors described the “modern view” of monocyte biology, but should also mention the different reactions, monocytes can undergo, such as the conversion into dendritic cells. In this regard it would also help if the mRNA expression pattern will give some information on transcripts that may play a role in monocyte differentiation (macrophages, dendritic cells, …) as opposed to the immune response. Are LPS-stimulated monocytes able to differentiate into different lineages?
  1. Analysis: Why did the authors not use monocytes directly after the isolation procedure without cultivation? IFN-γ was used as “reference” sample, what does this mean? It is an endogenous anti-viral protein. A more conclusive rationale to use two different agonists would be LPS as a PAMP and another agonist (e.g. HMGB1, histones or intrinsic nucleic acids) as a DAMP. The authors should justify their choice.
  1. One major drawback regards the missing protein data, at least for some examples, in order to evaluate whether the extreme mRNA transcript levels also reflect a comparable protein production or not. At least, examples from the literature concerned with this issue should be mentioned or discussed.
  1. Discussion: With their transcriptome analysis following monocyte stimulation with LPS the authors wanted to demonstrate the repertoire of genes transcribed under these in vitro conditions. Whether these conditions do reflect any in vivo situation should be taken with care and also be discussed, not only from the cell adhesive point of view, but also from the contribution of monocytes to the innate immune response. During the recruitment phase of immune cells from the circulation towards the site of injury/infection, the first line defense neutrophils are followed by monocytes. If these cells produce a series of cytokines, chemokines etc. as indicated here, which cells are targeted by these agonists (about one day following the tissue injury)? Are these agonists spread into the affected tissue site (like platelets eject their granule contents into the wound area for healing purposes), are they distributed into the blood stream (to sensitize other cells), or what happens? A clearer discussion on these aspects is required, and some literature study would help. Actually, another study to follow the actual major proteins (that are released from monocytes during the immune response) in their cell/tissue targeting would be an interesting question to follow.
  1. Minor: Several mistakes should be corrected, including misspellings and syntax errors, particularly singular and plural forms have been misused. The entire text should be checked for punctuation as well. A few examples are listed:

- Line 68: plural: act

- Line 70/71: LPS has already been mentioned as full wording.

- Line 77ff: Here, tissue factor may be mentioned shortly.

- Fig. 1: The abbreviation PE should be explained.

- Chapter 2.2: It is not entirely clear how many duplicates/triplicates were analyzed or whether only one sample for each transcriptomic analysis was assessed.

- Figs. 2-5: Labelling of x-axis (mRNA expression, x-fold, relative to ….) and y-axis (transcript) is missing. Also in Figs. 2-4: in the legends it is indicated that “the five non-induced samples are marked as individual (open) dots”, however, except for one, no dots are visible

- Line 159, wording: Influence of in vitro culture on the monocyte transcriptome.

- Line 173: No information on RNASE1 expression can be found in the tables.

- Line 176: The same is true for C1QA.

- Tables 1-11: All, not only a few, abbreviations of transcripts or proteins should be explained.

- Line 249: Among these five donors, we can observe …

- Line 271-273: a different font was used.

- Line 278: …, most likely, there is also …

- Line 333: Style of sentence is somehow strange.

- Line 342: Style of sentence: … that they predominantly function as sensitive …

- Line 345: singular: supports

- Line 347: …IL-8, a potent chemoattractant for neutrophils, …

- Line 352: delete: “without comparison”

- Lines 369 and 385: Delete the phrase “interesting” and leave this impression to the reader.

Author Response

Dear Reviewer,

We have here tried to address all your questions and comments.

  1. In the introduction we say that the monocytes change phenotype depending on tissue environment to become macrophage like cells of that tissue. However, the data we present here cannot give further indications to what these changes are. This would need the culturing in the tissue context and the results we present here is what responses monocytes can experience in the blood stream before entering the tissue. We are well aware of the tissue dependent changes that can be seen on tissue dwelling monocytes. However, it is outside the scope of this article. LPS in one of the most natural and common stimulators of immune cells as it is a common component of gram-negative bacteria and therefore a natural choice. It is also known that injection of LPS results in massive cytokine storm and can lead to death. We only look a short term effects in circulation and not in natural tissue surrounding.
  2. All the 0 values in this article is monocytes that not has been put in culture. This is also described in the last sentence in results section 2.1. We have also changed the text concerning reference and instead refer to as a non-bacterial inducer.
  3. Concerning transcript to protein we have now added a section in the discussion giving two new references describing a clear case on macrophages and secretome where transcriptome and proteome match very well but also an example from human lung mast cells where this is not the case. I shall here also mention the massive data we have on mouse and rat mast cells from our previous work where protein data on proteases and other granule components remarkably well correlate between RNA and protein levels. So, we are highly confident that the absolute majority of the changes in mRNA levels seen in this analysis also correspond to similar changes in protein levels.
  4. As mentioned above. The changes upon entering the tissue and what happens there is slightly outside of the scope of this article as the model system we here use can not give any information concerning such changes. I therefore think that at this point we cannot add any information of value for such a discussion even if that would have been great as it is a very interesting question- we fully agree.
  5. We have now addressed all the minor issues presented by the reviewer. All the changes are marked in red except when something is deleted.

Line 68 act has been changed.

Line 70/71 LPS description has been removed.

Line 77ff - we have added a sentence about tissue factor to the introduction.

Fig 1  PE has been explained

Chapter 2.2  There are only single runs per sample as we in previous studies using Ampliseq have seen that the results from multiple runs are almost identical.  See our previous addition to this special issue in IJMS on mouse peritoneal macrophages where we have duplicates.

Figs 2-5 We have now added Number of reads to the Y-axis and also changed the 0 time point values to open circles for better visibility in all four figures.

Line 159  We have now changed the wording and hope it reads better now. Also marked in red.

Line 173  and 176 Both values for RNASE1 and C1QA have been added to the tables.

Tables 1-11   Abbreviations to the transcripts have now been added to all transcripts that not are detailed described in the text such as MHC genes, cytokines chemokines and their receptors.

Line 249 text have been changed as suggested.

Line 271-273 font has been changed.

Line 278 wording and comma has been changed.

Line 333  Sentence have been changed hope it now reads better.

Line 342 Style of sentence have been changed

Line 345  supports have been changed

Line 347 comma has been added

Line 352 without comparison has been deleted.

Lines 369 and 385 interesting has been removed

Reviewer 3 Report

This study needs new rigorous experimental design to be deemed adequate for publication in a scientific journal. I am hard-pressed to see novelty throughout this study. Studying the inflammatory response in bulk-cellular population, with high levels of artificial agents is unlikely to reveal anything novel. This study did not. In addition, merely describing the changes in transcript levels is a weak experimental design. Weak experimental design is complemented by erroneous inferences. Your data does not demonstrate "a rapid protective response to high production of oxygen radicals, ........". A causal relationship is never established. 

Author Response

Dear Reviewer I must say that I am a little confused about your comments.

This is the first time a really quantitative analysis has been performed on human monocytes involving all 21 000 genes where we give actual transcript levels for five individuals of different age and sex.  We also present the actual data and not in the form of ¨Kinder Garten¨ red and blue Heat maps, which are so popular where all quantitative information has been lost. Transcript that differ by 3-4 order of magnitude in expression levels are in heat maps presented in the same red or blue color. It looks flashy but is almost totally useless for quantitative studies. We are honest and give actual values for the reader to appreciate.

Highly enriched bulk populations are also much better than single cell analysis by obvious reasons. There is very little RNA in single cells making it not possible to obtain good quantitative values. However, they are good for studying individual variation between cells and for lineage tracing. We are doing such studies on horse mast cells where we clearly can see the value of both techniques. However, single cell analysis is, by very obvious reasons, not good for quantitative studies. LC-MS is also not good for quantitative studies as they seem to be totally devoid of possibility to distinguish proteins present in 80 % or 0.001 % expression levels in a sample. We have sent purified proteins extracted from gels to three different MS labs and they send us a list or 100-200 proteins. When we ask them what is the major protein in the band that constitutes more than 80% of the material they ca not tell us. So LC-MS is not an alternative. It is a highly overrated technology that is much as possible should be avoided as it is almost impossible to trust any such data that has been published. Much of it is total crap and should be removed from Pubmed.

I am also very confused about the statement on artificial agents. LPS is purified directly from E. coli. E.coli is very natural it’s alive and kicking. And IFN-gamma is a recombinant protein, identical in primary structure to normal IFN-gamma and with the same biological activity and is used by almost all labs in the world studying its effects.  So what´s artificial?

Also confused about your comment on SOD2. We say that it indicates and how would you interpret the almost 30-fold induction of a protein involved in detox of oxygen radicals described in many articles in that context. Please tell me how you would interpret this result?

So what I ask you as reviewer -  what is the alternative and can you give us any article that is better than this study. I have screened Pubmed and cannot find any. I am sorry. Please send me such articles I would be very happy and tell me what alternative strategy we should have used to study the entire transcriptome by quantitative measurements except maybe 2.D gels where you extract all possibly 200 proteins that increase in expression level and performed immuno-histochemical analysis of them. Is this the alternative?  Now in the 21-century when we have highly reliable and validated transcriptome analysis methods. You can look at our previous papers on mast cell transcriptome and the validation by three independent technologies and all previous data from protein studies on mouse rat and human mast cells and see how well they correlate. So what is the alternative? Just give me a hint.

So please send me some better papers and I am open for an honest discussion if you can send me an email address so that I can communicate with you directly. We can even talk over phone if you prefer that medium.